# Beneficial Role of Exercise in the Modulation of *mdx* Muscle Plastic Remodeling and Oxidative Stress

**DOI:** 10.3390/antiox10040558

**Published:** 2021-04-03

**Authors:** Monica Frinchi, Giuseppe Morici, Giuseppa Mudó, Maria R. Bonsignore, Valentina Di Liberto

**Affiliations:** 1Department of Biomedicine, Neuroscience and Advanced Diagnostic (BIND), University of Palermo, 90134 Palermo, Italy; monica.frinchi@unipa.it (M.F.); giuseppe.morici@unipa.it (G.M.); 2Institute for Biomedical Research and Innovation, National Research Council, 90146 Palermo, Italy; marisa.bonsignore@irib.cnr.it; 3Department of Health Promotion Sciences Maternal and Infantile Care, Internal Medicine and Medical Specialties (PROMISE), University of Palermo, 90127 Palermo, Italy

**Keywords:** duchenne muscular dystrophy, training, treadmill running, swimming, voluntary exercise, muscle inflammation, ROS, antioxidants

## Abstract

Duchenne muscular dystrophy (DMD) is an X-linked recessive progressive lethal disorder caused by the lack of dystrophin, which determines myofibers mechanical instability, oxidative stress, inflammation, and susceptibility to contraction-induced injuries. Unfortunately, at present, there is no efficient therapy for DMD. Beyond several promising gene- and stem cells-based strategies under investigation, physical activity may represent a valid noninvasive therapeutic approach to slow down the progression of the pathology. However, ethical issues, the limited number of studies in humans and the lack of consistency of the investigated training interventions generate loss of consensus regarding their efficacy, leaving exercise prescription still questionable. By an accurate analysis of data about the effects of different protocol of exercise on muscles of *mdx* mice, the most widely-used pre-clinical model for DMD research, we found that low intensity exercise, especially in the form of low speed treadmill running, likely represents the most suitable exercise modality associated to beneficial effects on *mdx* muscle. This protocol of training reduces muscle oxidative stress, inflammation, and fibrosis process, and enhances muscle functionality, muscle regeneration, and hypertrophy. These conclusions can guide the design of appropriate studies on human, thereby providing new insights to translational therapeutic application of exercise to DMD patients.

## 1. Introduction

This review examines the role of exercise in the modulation of muscle plasticity and oxidative stress in the *mdx* mouse, the most widely used pre-clinical animal model for Duchenne muscular dystrophy (DMD), a fatal X-linked disorders characterized by progressive muscle weakness. Beyond the current medication, such as steroids, which show many side effects and can only slow down disease progression, and new gene- and stem cells-based strategies, still under experimentation [1,2], physical activity may represent an effective noninvasive therapeutic approach for DMD. However, if on one hand a moderate intensity exercise is usually associated to beneficial effects in healthy muscles [3], on the other hand the limited number of investigations in DMD patients and the lack of consensus regarding the fine balance between benefits of training and DMD muscle overuse and damage, impede the generation of definitive guidelines for exercise prescription [4,5].

It is well known that healthy skeletal muscle displays the ability to easily adapt to environmental stimuli, such as exercise and physical activity, through a plastic remodeling of the phenotype, strictly dependent on different properties of the applied stimuli. This adaptive remodeling, which plays an essential role in improving neuromuscular performance and/or enhancing endurance capabilities, includes metabolism modifications, structural changes, such as muscle pennation and fiber type switch, as well as size changes due to phenomena of hypertrophy [3,6,7]. The effects of exercise on muscle tissue affected by chronic disorders, such as DMD, have been extensively studied and reviewed [4,5,8,9,10,11,12]. However, since the beneficial effects of physical exercise depend on many parameters, such as type, duration, frequency, and intensity of training, and considering the large variation among individuals with regard to the magnitude of muscle remodeling in response to exercise, the lack of uniform and standardized protocols of training impairs comparisons between studies and translation of results obtained in animals to patients.

Therefore, in order to compare and shed some light on the many existing protocols of exercise, with the aim of identifying the best suited modality of training associated to beneficial effect on DMD muscle, here we present an updated data report about the effects of different protocols of physical exercise on muscles of *mdx* mice (Figure 1). The main modifications induced by exercise on *mdx* muscles, including the mechanism of fibrosis, the ability of satellite cells to regenerate damaged muscle tissue, the modifications of muscle performance (in term of endurance and strength) and structure (fiber type and muscle volume), as well as the metabolic processes associated with the control of fibers redox state and inflammation, will be reviewed in the next paragraphs. Furthermore, a difference in the functional response of *mdx* skeletal, cardiac and diaphragm muscles will be discussed, taking into considerations that the three types of muscle are not equally affected by the absence of dystrophin and exercise [13].

At the end of this discussion, we propose an optimized exercise protocol able to induce beneficial adaptive remodeling in *mdx* muscles, thereby providing new insights to translational therapeutic application of exercise to DMD patients.

## 2. Duchenne Muscular Dystrophy

DMD is an X-linked recessive progressive lethal disorder with a worldwide incidence of one in 5000 live male births [14]. It is caused by the lack of dystrophin, a critical muscle protein that connects the cytoskeleton of muscle fibers to the extracellular matrix, acting as an essential stabilizer of muscle fibers during contraction. Deletion of dystrophin, in both mature muscle fibers and myogenic stem cells, results in myofibers mechanical instability and susceptibility to contraction-induced injuries [15,16], as well as weakness, muscle loss [10], oxidative stress and inflammation [17,18]. The clinical course of the pathology is progressive and associated to life-long debilitation and shorter longevity: Patients with DMD first develop skeletal muscle weakness in early childhood, which quickly progresses to loss of muscle tissue, spinal curvature, paralysis, and premature death because of cardiorespiratory failure, usually in the third or fourth decade of life [1,14].

Muscle mechanical instability in DMD is associated with degeneration and regeneration of myofibers and activation of satellite muscle stem cells. Indeed, muscle injury activates satellite cells, which start to proliferate and to differentiate, leading, through a multistep process, to the formation of new regenerating fibers with centrally located nuclei, which represent a distinctive histological marker of DMD muscle [19,20]. The chronic nature of the pathology is associated at later stages to development of chronic inflammation, increased oxidative stress, inhibition of muscle fiber regeneration, depletion of the satellite cells pool and replacement of muscle by fibrotic and adipose tissue, and generating weakness in the diaphragm and limb muscles [21,22,23]. Importantly, dystrophic muscle fibers are more vulnerable to exercise-induced damage [24,25].

Unfortunately, at present, there is no efficient therapy for DMD. The classically available treatments, such as steroids, mostly interfere with inflammatory processes related to the pathology, thus reducing the immunological responses involved in the progression of the disease. However, these treatments show many adverse effects and can only slow down disease progression [1,26,27,28]. Indeed, research is still ongoing, and important developments have been achieved in the field, with the discovery of emerging therapies that are in the clinical trial phase or have already been approved. Briefly, gene-addition, exon-skipping, stop codon readthrough, and genome-editing-based therapies can restore the expression and the function of dystrophin, while stem cells-based therapy can partially replace damaged muscle tissue. Finally, other therapeutic approaches can improve muscle functionality by targeting pathways involved in the pathogenesis of DMD, such as inflammation and oxidative stress [1,2,29,30]. Although genome-editing based- and stem cells based-approaches have the potential to lead to meaningful life-changing therapeutic interventions, they show some limitations, especially in terms of cost burden and accessibility, and their therapeutic impact will be fully understood only in the next decades. Furthermore, genetic therapy for DMD patients remains an issue, taking into consideration the differences in mutations in the DMD gene and the complex mechanisms of gene expression regulation [31]. In addition to pharmacological, gene- and stem cells-based therapies, physical exercise, by inducing muscle plastic remodeling, represents a potential therapeutic approach for improving DMD patient outcomes and quality of life [5,9,32].

## 3. The *mdx* Model

The *mdx* mouse represents the most widely used pre-clinical animal model for DMD research [33,34,35]. *mdx* mouse was discovered in the early 1980s in a colony of C57BL/10ScSn, in which a spontaneous nonsense point mutation, consisting in a C-to-T transition at exon 23, determined loss of dystrophin, elevated serum creatine kinase (CK) and muscle damage [36]. Specifically, *mdx* skeletal muscle exhibits elevated myofiber necrosis, cellular infiltration, a wide range of myofiber sizes and several centrally nucleated regenerating myofibers. This phenotype is particularly evident in the diaphragm, which presents progressive degeneration, fibrosis and myofiber loss, resulting in significant strength reduction, thus closely reproducing the degenerative changes found in DMD muscles [33,34,35,37]. However, in general, although both *mdx* and DMD patients are genetically homologous and united by a complete absence of dystrophin, loss of dystrophin in *mdx* mice leads to a less severe phenotype, with minimal clinical symptoms. Indeed, the lifespan of *mdx* mice is only reduced by ~25% as compared to wild type (wt) animals, probably due to the activation of compensatory mechanisms, and/or to a species-specific feature of the muscle [38,39] and/or to the influence of the animal house environment, which hinders active movement and may therefore protect muscle from exercise-induced damage [40]. This experience has driven the development of new mouse models showing increased severity of the phenotype which better recapitulates the disease [33]. Nonetheless, the *mdx* mouse is still by far the most widely used animal model for DMD.

Several distinctive phases can be recognized in the pathology development in *mdx* mice [34,41]. In the first 2 weeks, even if higher mortality in *mdx* litters has been described, *mdx* muscle is still close to that of wt mice [42]. Muscle pathology becomes more pronounced between 2 and 8 weeks of age, when a clear presence of necrotic areas, newly regenerated centrally nucleated myofibers and high plasma concentrations of CK can be observed. In this phase, chronic inflammation, as evidenced by infiltration of inflammatory cells and high levels of inflammatory cytokines, can be also detected, together with fibrosis development [43,44,45]. However, the severity of fibrosis and loss of function in limb muscles is less in *mdx* mice than in human [46,47,48]. Severe dystrophic phenotypes, such as muscle loss, scoliosis, and heart failure, become evident in mice 15 months old or older, when a second robust phase of necrosis is not compensated by a regeneration process [49,50,51,52,53,54].

## 4. Effects of Exercise on Plastic Remodeling of *mdx* Muscle

Healthy striated muscles are characterized by a remarkable plasticity, as they can adjust the metabolic and contractile status in response to changes in functional demands. This adaptive remodeling, which plays an essential role in improving muscle performance, depends on the properties of the stimulus, especially in terms of intensity and duration of training [6,55]. Exercise intensity is classified accordingly to the physical exertion that the body uses when performing the activity [56] and is typically measured in metabolic equivalent task (MET), defined as the rate of energy expenditure at rest [57]. Exercise Activities with METs below 3 are considered as low intensity, with METs between 3 and 6 as moderate intensity, and with METs higher than 6 as high intensity [58].

Aerobic endurance exercise, characterized by repeated, sustained, low intensity contractions (i.e., long distance running, cycling and swimming) is generally associated to the switch of muscle fibers towards the type I, slow-twitch and fatigue-resistant [59], while resistance training, whose intensity is above 75% of the maximal capacity, characterized by low-frequency, high intensity contractions (i.e., body building and weight lifting), usually determines, accordingly to the protocol used, an increase in the proportion of type II fast-twitch myofibers and in force generation via muscle hypertrophy [55,60,61,62]. Generally, exercise provides numerous beneficial effects on skeletal muscle. However, since the effects of exercise engage several molecular and metabolic players, whose pattern of activation depends on the modalities of training, exercise might be also associated to detrimental effects in vulnerable DMD muscles, especially in terms of oxidative stress generation and fiber necrosis [3].

Similar to wt muscles, plastic remodeling of *mdx* muscle in response to exercise depends on many variables, including modality, intensity and duration of training. In rodent models, in order to study physiological adaptation associated with exercise, treadmill, swimming and wheel running exercise modalities have become quite popular [63]. While wheel running rely upon voluntary exercise, treadmill running and swimming can be controlled according to standardized protocols, especially in terms of duration and intensity set by the operator, thus allowing the investigator to track changes in the muscle response in relation to exercise parameters’ modifications. The adaptive remodeling induced by different typology of exercise in *mdx* muscles will be discussed in the next paragraphs.

### 4.1. Effects of Forced Running on Plastic Remodeling of mdx Muscle

Treadmill running represents the most common and effective modality of training used to investigate the effects of exercise on mouse muscles. Treadmill exercise intensity, expressed as the speed of running, can be set to low (<12 m/min), moderate (12–15 m/min), or high (>15 m/min) levels. Regarding exercise duration, although mice can run continuously for up to 2 h at a time, most studies used a 30–60 min/day training protocol, with a frequency ranging between 2 and 7 days/week, and a total duration lasting up to 24 weeks. When treadmill running is performed on an inclined platform, according to the negative or positive slope, it is referred as downhill or uphill running, respectively. Uphill running determines concentric muscle contraction and increases cardiovascular response, while downhill running represents a physiologically relevant model of eccentric loading associated to fiber membrane damage and subsequent metabolic adaptations [64,65]. Both training are associated muscle-specific increased workload as compared with running on a horizontal treadmill [8,66,67,68,69].

The main plastic remodeling induced by horizontal treadmill and downhill/uphill running exercise in *mdx* muscles have been summarized in Appendix A and Appendix A, respectively. This remodeling modulates muscle performance (in term of endurance and strength) and structure (fiber type and muscle volume), the mechanism of fibrosis, the degeneration/regeneration process, muscle metabolism together with muscle redox and inflammatory state. These last two parameters will be reviewed later.

Fibrosis, referred as the process of aberrant replacement of normal muscle tissue with connective tissue as response of reactive or reparative process, is a prominent pathological feature of skeletal muscle in patients with DMD, associated with severe muscle wasting and impairment of the muscle contractile functions and muscle regeneration ability [22,70,71,72]. Several studies have shown that treadmill exercise can efficiently modulate this process. Indeed, a low intensity exercise, at a speed of 9 m/min for two months, on a horizontal platform reduces collagen deposition in limb muscles of *mdx* mice [73,74,75]. In addition, low intensity running induced a strong beneficial effect on the degeneration-regeneration process of *mdx* muscle and a shift in favor of regeneration, both in limb muscles and in diaphragm [76,77], and reduced the percentage of necrotic area in *mdx* soleus and gastrocnemius muscles, while it seemed to exert opposite effects on *mdx* plantaris muscle [78]. These structural improvements are associated to the enhancement of muscle performances, such as increased grip strength, tetanic and specific force and resistance to fatigue [79].

On the contrary, higher intensity exercise, in the form of treadmill running at a speed equal or greater than 12 m/min or downhill/uphill running, independently of training duration, mostly enhanced collagen deposition and fibrosis area [15,16,80,81,82,83], and determined a series of detrimental effects on *mdx* muscles. Indeed, this type of exercise aberrantly up-regulated the phosphorylated form of extracellular signal-regulated kinase 1/2 (ERK1/2), p38 mitogen-activated protein kinases (p38 MAPK) and c-Jun N-terminal kinases 2 (JNK2), which might play a key role in the degeneration and regeneration process of *mdx* muscles [84]. The same type of exercise enhanced Ca^2+^ influx and sarcolemmal permeability [25,85], induced an increase in muscle injury/necrosis area [82,83,86,87,88,89,90,91,92,93,94], and promoted sarcoplasm fragmentation, oxidative stress and muscle inflammation [15,82,87,92]. Interestingly, a proteomic analysis revealed that 12 m/min treadmill running for 4 weeks failed to stimulate the metabolic changes associated to fast-to-slow transition, usually observed in aerobically trained muscles, decreased the expression of myosin regulatory light chain 2 and enhanced specific protein degradation, thus further reducing the amount of sarcomeres’ proteins in *mdx* mice [95]. All these molecular and histological changes are associated to deficit in muscle performances, such as the increase in fatigability and the reduction of forelimb strength [82,83,85,87,88,90,92,96,97,98,99], twitch and tetanic force [86]. In disagreement with all the other data about the effects of running on an inclined platform on muscle performance, 3 weeks of uphill running at a speed of 4 m/min initiated in three weeks old *mdx* mice increased soleus twitch tension and decreased muscle necrosis [100], likely due to the combination of low speed running and age of exercise initiation.

The equilibrium between degeneration and regeneration depends not only on the deposition of fibrotic tissue, but also on muscle stem cells functionality. Indeed, satellite cells impairment and, in particular, exhaustion and loss of stem cell properties seem to promote failure of the regeneration process and the subsequent degeneration overcoming [101,102]. Accordingly, stem cell-based therapies can represent a promising tool to induce DMD muscle regeneration [103,104]. Treadmill running seems to influence satellite cells properties of *mdx* muscles. Indeed, low speed treadmill running is associated to the increase in the regeneration area and to the reduction of connexin 39 (Cx39), a specific marker of injured muscle, in hind limb muscles [77], thus indirectly suggesting a beneficial effect of exercise on muscle stem cells functionality. Accordingly, 4 weeks of noninjurious isometric strength training, improved *mdx* phenotype and muscle performance, by inducing fiber hypertrophy, reducing fibrosis and increasing the number of satellite cells [105]. Furthermore, it has been demonstrated that treadmill running induced a telomere shortening in limb muscles in wt, but not in *mdx* mice, thereby suggesting that exercise might efficiently activate in dystrophic muscles compensatory mechanisms for this process [106], which has been hypothesized to contribute to the deficit of regenerative activity by promoting the premature senescence of satellite cells [107].

Based on the above reported experimental evidences, we can conclude that beneficial effects on *mdx* limb muscles induced by horizontal treadmill running exercise depend primarily on training intensity, being mainly associated to running speed ranging between 4 and 9 m/min. These positive effects, including fibrosis reduction, increased regeneration, improvement of muscle performances and structure, such as muscle hypertrophy and fiber type switch, seem to be not influenced by age, since they can be found in *mdx* mice 4–5 weeks old [108,109], 8 weeks old [74,76,77,110], and up to 20 weeks old [73,75,79], or by the duration of training, given the persistence of positive effects for exercise duration ranging between 1 and 6 months. On the contrary, horizontal treadmill exercise at higher intensity, with speed equal or greater than 12 m/min, or downhill/uphill running, generally exert detrimental effects on *mdx* muscles, and are often used to worsen the *mdx* phenotype before assessing the efficacy of drug treatment [8,81,82,84,87,92,99].

Interestingly, another key factor that influences exercise outcome is the number of training sessions. Indeed, while generally chronic exercise is associated to muscle beneficial effects, a single bout of treadmill running at a speed of 12 m/min or downhill running induced detrimental effects on *mdx* limb muscles, by generating membrane breakdown, myofibers necrosis, inflammation and oxidative stress [87,111,112,113,114,115,116]. Interestingly and consistently with *mdx* data, acute or single bouts of exercise resulted in increased serum CK activity and circulating myoglobin in DMD patients [117], likely due to the fact that the beneficial plastic muscle remodeling requires longer time to overwhelm muscle stress induced by acute exercise. In this context, another critical factor that seems to influence the outcome of exercise in *mdx* muscles is the sampling time following exercise. Indeed, it seems that tissue samples collected immediately after an acute o chronic high intensity treadmill running protocol is associated to increased damage, oxidative stress and inflammation as compared to samples collected 24 h or 96 h after the completion of the last training session [87], which may reflect a muscle adaptive response to exercise-dependent stress over time.

It is evident that exercise protocol might discordantly affect different muscles in *mdx* mice, mainly according to different muscle workload in response to training, and to specific muscle morphometric features. For example, as general principle, compared to the diaphragm, *mdx* limb muscles present larger necrosis area, but reduced fibrosis, following regeneration [13]. Furthermore, even skeletal limb muscles can differently respond to exercise, as, for example, low intensity training induced a decrease in gastrocnemius and soleus muscles necrotic area, while it exerted opposite effects on plantaris *mdx* muscle [78].

Similar to limb muscles, treadmill intensity strongly influences adaptive remodeling of cardiac muscle and diaphragm. Indeed, low intensity exercise, at a speed of 4–8 m/min on a treadmill, besides enhancing cardiac function, improved respiratory capacity [79] and increased the ratio between diaphragm regeneration and necrosis areas [76] without worsening cardiac and diaphragmatic fibrosis [79]. Conversely, higher intensity exercise seems to be associated to deleterious effects and enhancement of fibrotic tissue deposition in cardiac muscle and diaphragm. Indeed, running on a horizontal treadmill at 12 m/min induced a reduction in diaphragm twitch and tetanic tension [86], an increase in diaphragm necrosis [87] and an increase in cardiac collagen deposition and fibrosis [15]. Uphill running induced upregulation of phosphorylated p38 MAPK, phosphorylated ERK1/2 and calcineurin, extensive infiltration of inflammatory cells, together with increased interstitial fibrosis and adipose tissue in heart [118]. Similarly, downhill running increased the expression of Transforming Growth Factor Beta 1 (TGFβ1), a key modulatory of fibrosis deposition, on biceps brachii and heart [83], and enhanced diaphragm necrosis [25,93]. However, in contrast to these data, other papers have rather reported no detrimental effects to diaphragm muscle associated to running on a horizontal treadmill at 12 m/min [80,92,99].

Finally, diaphragm response to exercise seems to depend also on the sex of the subjects. Indeed, male *mdx* mice submitted to a 4-week period of 15–30 min running on a treadmill at a speed of 6 m/min, showed increased CK levels and inflammatory area as compared to trained females, suggesting that estrogens, whose receptor expression was increased following exercise in female *mdx* mice, may have contributed to the prevention of increased inflammatory process and diaphragm injury [119]. Accordingly, some investigations suggest that there are differences between female and male muscles, such as energy metabolism, fiber type composition, and contractile speed, greatly dependent on estrogen levels present in females [120]. Interestingly, estrogens may play an important protective role against muscle damage, by attenuating the inflammatory and oxidative process [121,122,123] and/or influencing muscle regeneration [124,125], thus affecting all the parameters modulated by exercise.

However, a similar protocol of training, consisting in 4 weeks of 30–60 min running on a treadmill at a speed of 4–4.8 m/min, did not seem to induce detrimental effects on male *mdx* diaphragm [76], thus suggesting the need for more specific comparative studies in order to address the impact of animal sex on training-induced muscle plastic remodeling. Nonetheless, it is worth emphasizing that the pre-training and the warm-up phase consisting of progressive increasing in muscle workload (in terms of intensity and duration of a single session of training) employed by Morici et al. [76] might have limited exercise-induced detrimental effects associated to diaphragm overload, promoting gradual plastic remodeling associated to morphological and functional improvement of male dystrophic muscle [76]. In support to the beneficial role of pre-training, it seems that even a treadmill exercise at a speed of 12 m/min was able to induce a reduction of serum CK levels if preceded by a warm-up period and a gradual increase of running speed [126], which is also responsible for the increased ability of *mdx* mice to complete a 30 min treadmill exercise session [87].

In conclusion, despite the limited available data on heart and diaphragm plastic remodeling in response to treadmill running complicate the definition of a reliable overview, they suggest that, similarly to limb muscles, beneficial effects on these muscles are mostly associated to low intensity treadmill training.

### 4.2. Effects of Swimming Exercise on Plastic Remodeling of mdx Muscle

Swimming training, although less extensively used in mice exercise interventions, presents some advantages compared to treadmill training. Indeed, swimming recruits all body muscles and ligaments, thus representing an effective form of aerobic endurance training. On the other hand, swimming represents a quite stressful modality of training, and requires careful and constant monitoring during the entire experiment to prevent mice from drowning or floating [67]. Swimming training intensity can be classified, according to the amount of swimming that takes place each day, as low intensity (20–59 min/day), moderate intensity (60–89 min/day), and high intensity (≥90 min/day) [67].

The effects of swimming on *mdx* muscle plastic remodeling have been summarized in Appendix A. Swimming, with variable duration up to 10 weeks, seems to produce adaptation to the functional demand and beneficial effects on *mdx* limb muscles, independently on intensity of exercise. Indeed, low intensity swimming for 30 min once a day, for 4 weeks, increased forelimb grip strength and decreased inflammation [127] and oxidative stress [128]. Interestingly, swimming exercise protracted for as long as possible, but never exceeding 25 min of training, once a day, for 10 weeks, exerted no detrimental effect even on 24 months old *mdx* mice, and increased relative tetanic tensions [129]. Even a more intense swimming exercise, executed for 2 h once a day for 15 weeks, but associated to a pre-training protocol, produced beneficial effects on limb muscles, by reducing muscle fatigability [130] and by increasing the proportion of oxidative fibers and twitch tension [130,131], confirming the beneficial effects of pre-training on *mdx* muscle adaptive response. Surprisingly, low intensity swimming consisting of 30 min bouts of training per day conducted on 4 consecutive days, seems to rather increase muscle necrosis and induce an increase in serum CK [132], maybe due to the short training duration which might prevent long-term beneficial muscle adaptation to training. Furthermore, a moderate protocol of swimming for 60 min once a day, for 2 months, produced detrimental effects on heart and diaphragm, exacerbating muscle degeneration, inflammation, and fibrosis in 11 months old *mdx* mice [133]. These data suggest that, similar to the effects of treadmill training, limb muscles, heart and diaphragm do not show the same adaptive remodeling in response to swimming training. Interestingly, a single 20 min swimming session seems to be detrimental for *mdx* muscle by inducing a membrane breakdown in tibialis anterior muscle [134], thereby confirming that acute exercise is generally associated to deleterious muscle remodeling.

### 4.3. Effects of Voluntary Running on Plastic Remodeling of mdx Muscle

Different to forced running, voluntary wheel running allows mice to exercise at a lower intensity and to freely run. Although this training may generate individual differences in the amount of exercise among mice, the animals are subjected to a lower stress [67]. The effects of voluntary running on *mdx* muscle plastic remodeling have been summarized in Appendix A. Voluntary exercise, independently of training duration, seems to be generally associated to beneficial effects on the functional properties of *mdx* muscles. Indeed, short-term (1 week) voluntary wheel running, initiated when mice were 2-4 months old, by activating calcineurin pathway and modifying the expression program of genes involved in excitability and slower contractile phenotype, preserved muscle excitability and improved tibialis anterior muscle fragility in *mdx* mice, without worsening weakness [135]. Longer periods of voluntary exercise seem to positively affect *mdx* muscle physiology as well. For example, 3 weeks of voluntary wheel running in maturing 21 days old *mdx* mice was not detrimental and enhanced endurance capacity by inducing molecular adaptions in both skeletal and cardiac muscle, such as increased total contractile protein content and markers of aerobic metabolism [136]. Four weeks of voluntary wheel running initiated when mice were 2 months old, increased cross sectional area and reduced protein ubiquitination in triceps brachialis muscle [137], while 8–9 weeks of voluntary wheel running, initiated at the age of 3–4 weeks, improved muscle performance and induced adaptive response in *mdx* muscles, such as increased muscle mass in soleus and the shift of fibers toward a less fatigable phenotype [138,139], without improving the activity of mitochondrial enzymes of the Krebs cycle, β-oxidation, and the electron transport chain [138]. This nonadaptive mitochondrial response is unusual considering that adaptive myosin heavy chain isoform response and the shift from IIb toward IIa fibers typically occurs together with the enhancement of mitochondrial enzyme activities.

Continuing with the review of positive effects associated to voluntary exercise, 12–18 weeks of low resistance wheel running initiated in 4 weeks old *mdx* mice led to increased utrophin, a dystrophin protein homologue that enhances dystrophic muscle function [140], improved contractile function and reduced fatigability, and increased aerobic metabolism along with a trend of fiber type transformation toward a slower-contracting and less fatigable fiber type [141,142,143]. Beneficial results were also obtained with 12 weeks of voluntary progressive resistance wheel running, which was able to increase forelimb muscles performance in 4–5 weeks old *mdx* mice [144]. Noteworthy, 3 or 11 months of wheel running seem to exert positive effects on *mdx* limb muscles even when initiated at an age of 6 or 7 months, by ameliorating the age-associated loss in tension production and fatigability [145,146]. Finally, 1 year of voluntary wheel running induced positive exercise-induced remodeling of limb muscles, such as the increase in the muscle mass and tetanic force [147].

In addition to this large body of evidence, mostly demonstrating beneficial effects of voluntary exercise on dystrophic limb muscle, other studies have rather described detrimental roles associated to this type of training. Indeed, 2–3 weeks of voluntary wheel running started in 9–12 weeks old mice exacerbated limb *mdx* phenotype and increased the amount of damaged necrotic tissue and macrophage infiltration, associated to a more pronounced inflammation and fibrosis genes dysregulation [148,149,150]. Moreover, wheel running for 8 weeks increased skeletal muscle fibers and endothelial cells apoptosis in young *mdx* mice [151], although it remains controversial whether this process is related to repair/regeneration mechanisms or may contribute to the pathogenesis of muscle damage [152].

Based on the above reported literature data, it is possible to conclude that voluntary wheel running seems to be definitely associated to beneficial effects on *mdx* limb muscles especially if initiated early, at an age ranging between 1 and 6 weeks, and independently on the duration of training. On the contrary, under the same conditions of training duration, a later initiation of voluntary exercise is associated to discordant outcomes, either beneficial [135,145,146] or detrimental, such as the enhancement of inflammation, necrosis and fibrosis [148,149,150,153]. These differences, in addition to be influenced by changes in the age and the subsequent stage of the dystrophic process, which may affect muscle regeneration and remodeling ability [154], are likely dependent on the large individual variations associated to this typology of exercise. Furthermore, as already pointed in relation to other modalities of training, sex of studied animals and the presence of estrogen receptors may contribute to the prevention of muscle injury in *mdx* mice and affect muscle response to exercise [119]. Accordingly, voluntary activity was not detrimental for *mdx* limb and cardiac muscles of either sex but increased absolute maximal force and muscle weight only in female mice [146].

Independently of age of exercise initiation, another factor that strongly influences voluntary exercise outcome is the number of training session. Indeed, a single bout of training seems to be detrimental and induces muscle damage, fiber apoptosis [152,155,156,157], and an increase in membrane permeability [157], thereby confirming acute deleterious effects associate to training.

Like treadmill running and swimming, cardiac and diaphragm muscles, as compared to limb muscles, need different generalizations. Indeed, for example, while long term wheel running, initiated when mice were 4 weeks of age, improved cardiac and plantarflexor function in the *mdx* mouse, it greatly impaired diaphragm function, likely due to increased respiratory muscle workload and frequency of eccentric contractions during exercise training [147]. However, other studies have rather demonstrated that 1 year of wheel running in young *mdx* mice increased diaphragm active tension [139], while lifetime wheel running increased its contraction time [158], thereby decreasing progression of muscular dystrophy. A large variation in muscle response to voluntary training can be also observed in cardiac muscle. For example, 12 weeks of low resistance wheel running initiated in 4 weeks old *mdx* mice led to the increase in heart mass [141], while a similar protocol of training in age-matched *mdx* mice did not seem to produce variation in cardiac relative mass [144]. Furthermore, 3 or 8 weeks of voluntary wheel running did not exacerbate heart *mdx* phenotype in aged [159] and maturing *mdx* mice [136], respectively, and, in the latter case, enhanced the expressions of aerobic metabolism markers, such as citrate synthase (CS) and β-hydroxyacyl-CoA dehydrogenase activities [136]. On the contrary, 4 to 4.5 months of voluntary wheel running, initiated when mice were 4–5 weeks old, reduced left ventricular function [143]. Finally, 4 weeks of voluntary wheel running accelerated the progression of ventricular dilation and fibrosis in 7 weeks old *mdx* mice [160], while a longer training period in 7 months old mice, did not affect left ventricular function, structural heart dimensions, cardiac gene expression of inflammation, fibrosis, or remodeling markers [146].

In conclusion, although experimental evidence has demonstrated muscle beneficial effects induced by voluntary exercise, especially in limb muscles, the large individual variation in terms of effective muscle load associated to this modality of training impedes a clear correlation between exercise and observed effects and hinders an exercise protocol standardization [67,148], thus suggesting that this modality of training might not represent the most efficient therapeutic approach for DMD patients.

## 5. Mitochondria Impairment and Redox Equilibrium in DMD Muscles

Loss of dystrophin in DMD muscles generates a systemic metabolic impairment, which is a key contributor to the etiology of the disease. Indeed, DMD is characterized by a significant dysregulation of intracellular Ca^2+^ homeostasis and by a deficiency in glycolysis, purine nucleotide, and tricarboxylic acid cycle, as well as in oxidative phosphorylation [161,162,163,164,165,166,167,168,169], which result in bioenergetic impairment and reduced ATP levels [170,171]. This, in turn, leads to the impairment of muscle contraction and, ultimately, to cell necrosis.

Muscle necrosis is closely associated with increased inflammation and oxidative stress [172], and both these processes can be efficiently modulated by exercise.

Oxidative stress is a cell condition caused by an excess of intracellular free radicals, such as reactive oxygen species (ROS) and radical nitrogen species (RNS). Normally, the levels of free radicals inside the muscle fibers are strictly balanced by controlling the rate of their synthesis and removal by antioxidants defenses. Usually ROS are physiologically produced at low amount in healthy cells and an accurate and well organized control of their levels guarantees the maintenance of physiological cell functions involving ROS as cell signaling molecules, i.e., control of gene expression, regulation of cell signaling pathways and modulation of skeletal muscle force production [173,174,175]. On the other hand, excessive ROS/RNS production and/or defects of antioxidant systems may impair the cellular redox balance, inducing oxidative stress in cells, subsequent damage to biologic macromolecules, and cell death [176,177].

A large body of evidence shows that oxidative stress, driven by free radicals produced in damaged myofibers or released by inflammatory infiltrating cells, represents a key pathogenic event in DMD [18,172,178,179,180,181]. Accordingly, antioxidant drugs (coenzyme Q_10_, green tea extracts, resveratrol, *N*-acetylcysteine), although with several caveats, are considered a promising treatment strategy in muscular dystrophy [182,183]. The prominent role of oxidative stress in the pathology has been suggested quite early by observing that muscles from DMD patients showed higher level of lipid peroxidation and enhanced antioxidant enzymes activity [184]. In general, abnormal enzymatic antioxidant responses and increased levels of oxidant molecules and markers of oxidative damage to macromolecules, such as 8-hydroxy-2′-deoxyguanosine, oxidized proteins, and lipids, have been found in DMD muscles and blood [185,186,187,188,189]. Interestingly, the extent of oxidative damage further increased with pathology progression and aging of affected subjects [189].

Mitochondria represent a crucial site of ROS production, and impairment of mitochondrial respiration strongly contributes to DMD pathogenesis in both patients and animal models [164,190,191]. Specifically, DMD-induced perturbation of intracellular Ca^2+^ homeostasis, metabolism and ATP production seems to be bidirectionally associated with mitochondria impairment [165,179,190]. Under physiological conditions, the mitochondrial electron transport chain transfers a single electron to molecular oxygen leading to the synthesis of ATP and superoxide ions that can be neutralized by antioxidant systems [192]. Loss of dystrophin in DMD induces destabilization of cell membrane and disruption of cytoskeleton organization, and the subsequent abnormal Ca^2+^ influx, which activates proteases and leads to mitochondrial Ca^2+^ overload and dysfunction [193,194]. Specifically, Ca^2+^ overload triggers mitochondrial permeability transition pore (PTP) opening, alteration in mitochondrial membrane potential and swelling of the organelles, ROS overproduction, and oxidative stress generation, which result in the subsequent exacerbation of mitochondrial dysfunction that further impairs cell bioenergetics, Ca^2+^ homeostasis and cell viability [185,195,196]. In other words, loss of dystrophin generates a feed-forward loop in which mitochondria impairment induces muscle ROS increase and oxidative stress, which further compromise mitochondria functions, leading to even greater production of ROS and muscle necrosis [197]. Furthermore, monoamine oxidase (MAO), a mitochondria enzyme catalyzing the oxidative deamination of biogenic amines to generate H_2_O_2_, is increased in *mdx* muscles [198], contributing to the overall oxidative imbalance by generating higher levels of H_2_O_2_, which in turn alter the redox homeostasis and causes myofibrillar protein oxidation, muscle damage and impairment of contractile function [198]. In addition to the alteration in mitochondria enzymatic activity, reduced density of sub-sarcolemmal mitochondria and abnormal localization of inter-myofibrillar mitochondria contribute to the overall mitochondria dysfunction in *mdx* mice [191]. Notable, elevated levels of mitochondria H_2_O_2_ in *mdx* mice, due to impaired oxidative phosphorylation, aberrant MAO expression and activity and altered mitochondria biogenesis and dynamics, occur very early in the disease process in 4 weeks old mice, before the onset of the disease and myofiber necrosis, and are associated with the induction of mitochondrial-linked caspase 9, necrosis, and severe myopathy in skeletal muscles and diaphragm [166,199,200]. These pieces of evidence indicate that mitochondria dysfunction is likely involved in the initial phases of pathology, being an essential contributor to DMD pathogenesis [178]. Accordingly, genetic or pharmacological overexpression of Peroxisome proliferator-activated receptor gamma coactivator 1-alpha (PGC-1α), a master regulator of mitochondria functions, ameliorates *mdx* mice phenotype [201,202,203].

Besides mitochondria, NADPH oxidase (NOX), phospholipase A2 (PLA2), and xanthine oxidase (XO) are other important muscle sources of ROS in DMD. NOX enzymes transport electrons across biological membranes to reduce oxygen to superoxide or H_2_O_2_ [204,205]. Skeletal muscles express three isoforms of NOX (NOX1, NOX2, and NOX4) that have been described as key regulators of redox homeostasis [206,207]. In particular, NOX2 represents the major source of skeletal muscle ROS during contractions, while NOX4 is mostly involved in skeletal muscle hypertrophy induced by muscle overload [206]. Indeed, muscle contraction induces the association of NOX2 regulatory subunits, leading to its activation and the subsequent ROS generation [208]. NOX2 is also involved in the regulation of excitation-contraction coupling, since moderate levels of superoxide anion produced by its activity in the sarcoplasmic reticulum (SR) can activate ryanodine receptors and regulate the proper release of calcium from intracellular stores [209,210]. PLA2 stimulates NOX activity [211] and cleaves arachidonic acid, a lipoxygenases substrate for ROS production, from phospholipids contained in membranes, both in resting conditions and during contraction [212]. XO is a cytosolic enzyme that catalyzes the hydroxylation of hypoxanthine to xanthine and of xanthine to uric acid, generating ROS [213]. Several pieces of evidence have demonstrated a large enhancement of NOX-dependent ROS production in dystrophin-deficient heart and skeletal muscle [214], together with a hyper-activation of XO enzyme [215]. Accordingly, pharmacological inhibition of NOX and XO activity significantly ameliorated *mdx* muscle function by protecting from tissue damage and inflammation [215,216].

Importantly, membrane damage during DMD and subsequent muscle necrosis activate resident mast cells, which in turn secrete inflammatory mediators to recruit circulating inflammatory cells from the surrounding vasculature, like neutrophils and macrophages, which contribute to ROS generation to promote phagocytosis [217]. However, the chronic inflammatory state of DMD muscle leads to excessive ROS generation, oxidative stress exacerbation and to secondary damage of previously uninjured fibers [218,219].

In summary, inflammatory cells, mitochondria, NOX, and XO are the main sources of ROS in DMD [219,220]. Although mitochondria in muscle fibers have traditionally been considered the major intracellular generator of ROS, many recent evidences have suggested NOXs as the most significant source in DMD muscles, especially during contractions, since the increase in cytosolic ROS is quicker and greater than the rise in mitochondrial ROS [207,219,221]. In support to this thesis, pharmacologic o genetic blockade of NOX activity inhibits or attenuates contraction-induced increase in ROS [222,223,224].

In order to counteract the excessive increase in radicals and oxidative stress, muscle cells arrange enzymatic and non-enzymatic defenses. The primary enzymes in cells are represented by superoxide dismutase (SOD), glutathione peroxidase (GPX), and catalase. Other antioxidant enzymes, such as peroxiredoxins, glutaredoxins, and thioredoxin reductases, and non-enzymatic antioxidants (e.g., glutathione, uric acid, and bilirubin) also contribute to cellular protection against oxidation [225]. SODs catalyze the dismutation of superoxide in molecular oxygen and hydrogen peroxide [226], which, in turn, is converted into water by catalase or GPX. Three isoforms of SOD (SOD1, SOD2, and SOD3) were identified in mammalian cells. SOD1, which uses copper-zinc as a cofactor, is mainly located in the cytosol. SOD2 requires manganese as a cofactor and is localized in the mitochondria, while SOD3, which contains copper-zinc as a cofactor, is mainly located in the extracellular space [174]. In skeletal muscle fibers, SOD isoforms are involved in contraction of myogenic fibre [219] and their activity is greatly dependent on muscle fiber type and modality of exercise. Specifically, slow twitch and Type IIa fibers have higher levels of SOD2 and SOD1 than Type IIb and IIx fibers [226,227].

Similar to SOD, Activity of GPX and catalase changes according to fiber types, being highest in skeletal muscles composed of highly oxidative fibers compared to fibers with lower mitochondrial content [174,175]. Interestingly, muscles from *mdx* mice exhibit increased expression of antioxidant enzymes, such as SOD1, SOD2, GPX, and catalase, in the pre-necrotic state, indicative of a cellular response to oxidative stress [178,219], although other studies have provided contrasting results about the levels of antioxidant enzyme activities in dystrophin-deficient muscle, likely due to the different stage progression of the pathology [184,228,229,230]. On the contrary, decreased levels of glutathione, accompanied by a concomitant increase in the activity of glutathione metabolizing enzymes (GPX and glutathione reductase), have been detected in DMD muscles, contributing to the generation of oxidative stress in these muscles [181,231].

Finally, the overall control of fibers redox state is also dependent on SR and on its role in protein-folding homeostasis [232,233]. Indeed, the accumulation of unfolded or misfolded proteins within the SR leads to SR stress and to the activation of an adaptive complex intracellular signal transduction pathway aimed to resolve protein misfolding and re-establish SR proteostasis through autophagy and enhancement of nuclear factor E2-related factor 2 (Nrf2)-dependent gene transcription [17,234,235], involved in mitochondrial biogenesis [236] and in the upregulation of antioxidant enzymes in response to oxidative stress [237,238]. However, depending on the severity of SR stress, activation of alternative pathways can even enhance ROS production, leading to further oxidative stress, and, eventually, apoptosis [239]. It has been reported that skeletal muscles in *mdx* mice are more susceptible to oxidative stress compared to those of wt animals [240], and that this difference might be, at least in part, attributed to the deregulation of Nrf2 signaling in *mdx* muscles [241]. Furthermore, SR stress markers, such as caspase-12, are increased in dystrophic muscle of *mdx* mice and DMD patients, suggesting that dystrophin deficiency leads to disruption of muscle SR homeostasis, which contributes to the worsening of oxidative stress, inflammation and DMD phenotype [242]. Accordingly, SR stress inhibitors restore mitochondria functions, mitochondrial Ca^2+^ uptake, and improve contractility of the diaphragm in *mdx* mice [243].

## 6. Exercise Modulation of *mdx* Muscle Oxidative Stress and Inflammation

It is known that muscle cells can generate ROS in resting conditions, and even more during exercise [244,245]. Indeed, during exercise, the increased energy demand leads to the boost in mitochondrial activity and oxygen consumption and subsequent muscle ROS generation [212]. The exercise-induced increase in ROS levels is essential to modulate muscle adaptation and force production, and exerts beneficial or detrimental effects according to ROS concentration, which mostly depends on duration and intensity of exposure and training status of the subject [174,205,245].

Usually, endurance training produces low concentrations of ROS, which in turn induce the expression of antioxidant enzymes and other defense mechanisms [226,246] and beneficial muscle adaptations [247,248]. Noteworthy, the beneficial effects of exercise are inhibited by administration of antioxidant compounds, such as vitamin C and E [249,250,251] thus suggesting that ROS, at moderate levels, act as critical signals in exercise promoting useful plastic adaptations in muscle. Endurance training compensates also for the decrease, during aging, PGC-1α pathway [252], the master transcriptional coactivator involved in the regulation of mitochondrial biogenesis and metabolism. Specifically, exercise-induced PGC-1α expression strongly improves the control of energy metabolism by promoting mitochondrial oxidative metabolism and mitochondrial biogenesis, protects against development of sarcopenia, muscle oxidative stress, and inflammation, and drives angiogenesis [203,253]. In addition, exercise, by activating several interconnected intracellular signaling, including PGC-1α and AMP-activated protein kinase (AMPK) pathways, in aged muscles, ameliorates mitochondrial quality by altering mitochondrial dynamics [254,255,256] and promoting mitophagy of damaged or dysfunctional mitochondria [257,258]. All the above mentioned processes contribute to guarantee muscle mass and strength preservation in senior sportsmen compared to age-matched sedentary subjects [259]. These beneficial effects of exercise on aged muscles strongly support the potential positive effect of low intensity training on DMD patients, which show progressive muscle weakness and deterioration with age [1,54]. Indeed, although the shorter longevity, DMD muscles share several features with aged muscles, such as increased oxidative stress, fibrosis, sarcopenia, altered properties of satellite cells and impaired regeneration [260], which might be efficiently compensated by low intensity training. Accordingly, increasing, for example, PGC-1α expression (by exercise, pharmacologic agents, genetic manipulation, etc.) exerts beneficial effects on aged muscles as well as on DMD muscles [203,261].

In order to regulate the overall muscle redox state, exercise can also influence activity and expression of antioxidants systems. In details, regular/moderate exercise enhances antioxidant defenses by inducing the activity of endogenous antioxidant enzymes such as SOD, GPX and catalase [262,263]. Moderate exercise, indeed, promotes the activation of Mitogen-activated protein kinase (MAPK) pathway, which in turn increases the expression of antioxidant enzymes, thus counteracting excessive ROS generation and stimulating adaptation to exercise [264]. Specifically, although some studies reported that prolonged endurance exercise training does not enhance muscle SOD activity [225,246], most investigations have shown that endurance exercise increases SOD muscle activity [227,265,266,267,268,269] proportionally to the intensity and duration of exercise [227,267]. Specifically, endurance training, or moderate exercise, seem to primarily induce the enhancement of SOD2 protein and activity in type IIa fibers [226,246,270,271], even if several reports have also described SOD1 activation [272]. More recently, the increased expression of SOD3 mRNA has been also reported in skeletal muscle after acute exercise [273].

While is still controversial whether catalase expression in skeletal muscle is enhanced by chronic exercise [225,227,268], GPX, similarly to SOD, is strongly increased in muscle fibers during training [246,265,274,275,276,277], according to both intensity and duration of exercise. Specifically, high intensity exercise induces a greater increase in muscle GPX activity if compared to low intensity exercise [227].

Similarly to wt muscle, low intensity exercise seems to be able to counteract oxidative stress in *mdx* muscles. Indeed, 6 months of low intensity treadmill running increased PGC-1α expression in *mdx* cardiac muscle [79], which acts as a central inducer of mitochondrial biogenesis and a primary regulator of redox balance inside contracting muscle [253,278]. 8 weeks of low intensity treadmill running, at a speed of 9 m/min, started when mice were 4 weeks old, reversed lipid and protein oxidation in the gastrocnemius of *mdx* mice [108,109], increased the antioxidant activity of glutathione, free thiols concentrations and reversed the changes in mitochondrial respiratory chain complex activity associated to the disease, thus improving energy metabolism [108]. The same exercise protocol, started when mice were 11 weeks old, although associated with higher levels of lipid peroxidation and protein carbonylation in the tibial anterior and gastrocnemius muscles, was also able to enhance SOD activity and total antioxidant capacity [73]. These observations suggest that the exercise-dependent increase in antioxidant enzymes is not sufficient to reverse oxidative damage to macromolecules, likely due to the age of treatment initiation, but can sustain the attenuation of intramuscular collagen fibers deposition [73], which occurs in *mdx* mice after the tenth week of age [8]. A modulation of antioxidant defenses was also observed following 30 days of low intensity endurance exercise on rotating treadmill, which produced an increase in quadriceps SOD1 levels [110]. In agreement with the beneficial effects associated to exercise and dependent on the up-regulation of antioxidant systems, the catalytic mimetic of SOD and catalase or catalase overexpression have been shown to reduce muscle damage and markers of oxidative stress [279] and ameliorate functions of skeletal muscles in *mdx* mice [280].

Different protocols of low intensity training, such as 30 min of swimming per day for 4 weeks, corroborated the positive effects of training on *mdx* oxidative stress, showing a decrease in protein carbonylation in the gastrocnemius muscle when compared with the non-exercised *mdx* group [128]. The only exception to positive effect associated to low intensity training is represented by the study of Faist et al., in which running on a treadmill at a speed of 8 m/min, even in presence of a pre-training, decreased oxygen consumption, respiratory control indices and GPX activity and increased lipid peroxidation in young (4 weeks) *mdx* muscle [281]. Surprisingly, no effects or even opposite effects were obtained by the same protocol of exercise in adult (16 weeks), suggesting that young *mdx* muscle is more vulnerable to exercise-induced to oxidative stress as compared to adult muscle, likely due to the higher degree of regenerating fibers in adult *mdx* mice [281]. However, as already described, other studies have shown beneficial effects associated to a similar protocol of exercise in young *mdx* mice [108,109], thus suggesting the need for further investigations finalized at a more reliable standardization of a training protocol associated to a reduction of DMD muscle oxidative stress.

The role of voluntary exercise on *mdx* oxidative stress regulation is controversial. On one hand, 7 weeks of voluntary run in 6 to 7 weeks old *mdx* mice enhanced autophagy and reversed the oxidative phenotype by counteracting the decrease of PGC1-α content [282], whose increased activation has been shown to ameliorate DMD pathology [202,283]. On the other hand, the same protocol of exercise, performed by age-matched *mdx* mice, increased oxidized glutathione and protein carbonyl levels in gastrocnemius, without increasing ER stress [284]. One week of voluntary wheel running promoted the slower contractile phenotype switch in tibialis anterior muscle of 2–4 months old *mdx* mice and up-regulated the expression of Mitochondrial transcription factor A (Tfam), Nuclear respiratory factor 1 *(*Nrf1), and peroxisome-proliferator-activated receptor-gamma co-activator 1 beta (PGC1-β) [135], which are important regulators of the oxidative metabolism [285,286]. Three weeks of voluntary wheel running in 21 days old *mdx* mice increased serum antioxidant capacity and markers of oxidative metabolism, such as CS and β-oxidation activity, necessary to determine metabolic adaptations to endurance exercise, in both quadriceps and heart muscles [136]. Similar metabolic modifications were also observed in the gastrocnemius of 4 weeks old *mdx* mice submitted to 12 weeks of voluntary wheel running together with the up-regulation in cytochrome c oxidase subunit 4 levels and a trend PGC-1α increase [141], which suggest enhanced mitochondria metabolism and biogenesis in response to training. However, given the limited amount of available data and the large variability associated to voluntary training, further studies are necessary to clarify the role of this type of exercise on *mdx* oxidative stress.

In healthy muscle cells, when ROS amount reaches a maximum peak, the linear relationship between ROS and muscle generated force fails and a further increase in ROS induces a strong reduction in the force and muscle oxidative damage, such as protein carbonylation, DNA damage, and RNA oxidation [287]. Usually, oxidative damage and inflammatory processes in active myofibers are associated to prolonged and intense exercise [288,289,290]. One of the main mechanism involved in oxidative damage to muscle fibers following strenuous exercise, which is associated to ROS production, proinflammatory cytokines and increased Nuclear factor-kB (NF-kB) activation, is the impairment of pathways involved in maintaining mitochondrial integrity and morphology [291,292]. Moreover, excessive ROS production is associated with the impairment of PGC-1α pathway and alters calcium homeostasis, leading to aberrant NF-kB transcriptional activity, activation of proteolytic systems and muscle wasting [293]. Finally, during intense prolonged exercise, XO activity is significantly increased leading to the excessive ROS generation, oxidative stress and muscle damage [294], while a single session of exhaustive exercise is able to determine oxidative damage only in untrained persons or senescent muscle, due to a higher vulnerability to oxidative stress [295]. In addition to these intracellular sources, ROS can also be produced from non-muscle sources during training, especially by immune cells activated by exercise-elicited muscle injuries [296,297], and by oxidation of catecholamines, whose plasmatic concentrations increase in response to exercise [298].

Similar to wt muscle, high intensity exercise is mostly associated to worsening of *mdx* phenotype and muscle redox status. Indeed, acute downhill treadmill running at a speed of 15 m/min caused significant increases of XO activity and the subsequent systemic ROS generation as indicated by elevated urinary isoxanthopterin in *mdx* mice [215]. Four weeks of moderate intensity treadmill running at 12 m/min speed increased ROS [92] and oxidized glutathione levels [15], and further worsened calcium homeostasis and sarcolemmal permeability [85]. Similarly, 6 weeks of the same protocol of exercise increased protein thiol oxidation [111], which was also significantly enhanced even after a single treadmill session [87]. Finally, 12 weeks of treadmill running at 12 m/min induced a significant reduction in PGC-1α expression, accompanied by a marked downregulation of Sirtuin 1 (Sirt1) and Peroxisome proliferator-activated receptor γ (Pparγ) [88], which play key roles in exercise-induced adaptation, via stimulation of the fast-to-slow transition, mitochondrial biogenesis and cellular antioxidant response [283,299]. The same exercise is associated also to a reduction of the autophagy marker BCL2 Interacting Protein 3 (Bnip3) [88]. Interestingly, a single session of the same protocol of exercise executed by 16 weeks *mdx* mice did not influence the expression of PGC-1α, Sirt1 and Pparγ [88], in contrast to all the other data on muscle plastic remodeling demonstrating mostly detrimental effects associated to acute exercise [87,111,112,113,114,115,116,134,152,157].

It is well known that inflammation process is strictly linked to oxidative stress. Indeed, oxidant species can mediate activation of NF-kB [300] which in turn regulates the expression of genes involved in the inflammatory and stress response [301]. Conversely, inflammation leads to immune cells infiltration in damaged muscled, which in turn determine the release of free radicals, generation of oxidative stress and chronic inflammation, thereby forming a self-propelled vicious cycle [302].

Inflammation is an early event in the pathological process of DMD, closely associated with myonecrosis, already evident in 3–4 weeks old *mdx* mice [303]. Indeed, loss of dystrophin leads to a fragility of plasma membrane that is easily injured during muscle contraction, inducing extracellular calcium influx, which in turn activates the NF-kB inflammatory pathway and enhance inflammatory cytokine release, e.g., Tumor Necrosis Factor α (TNFα) and Interleukin (IL)-1β, which further impair muscle regeneration [304]. Hence, activation of NF-kB is an important mediator of muscle damage and DMD pathology in both humans [305] and *mdx* mice [303,305,306]. Accordingly, calcium influx is sufficient to induce muscular dystrophy [307], while genetic ablation [306] or pharmacological blockade [305,308,309] of NF-kB alleviate the severity of the disease and improves muscle phenotype of *mdx* mice. Glucocorticoid therapy interferes with inflammatory processes related to the pathology and mitigates elevation of NF-kB and oxidative stress, determining the subsequent reduction of fiber damage and the enhancement of muscle functional properties [151,310]. However, this treatment is associated to severe side effects such as weight gain, behavioral changes, growth issue, late puberty, bone demineralization, and gastroesophageal reflux, and can only slow down disease progression [26,27,28,311].

Interestingly, emerging evidence described the beneficial effects induced by exercise on *mdx* inflammatory processes. In details, low intensity swimming training decreased Monocyte chemoattractant protein-1 (a chemoattracting agent playing a crucial role in the coordination of the inflammatory response) levels and, although increased CD68^+^ macrophage numbers, shifted the macrophage population from to M1 pro-inflammatory type to the regenerative M2 type, thus inducing a modulation of inflammatory state of gastrocnemius, a switch of fibers to the oxidative slow type and a grip strength increase [127]. A low intensity running exercise on treadmill, preceded by a pre-training, exerted anti-inflammatory effects on *mdx* quadriceps muscles by reducing NF-kB expression [76] and increased serum adiponectin, which shows anti-inflammatory properties as well as modulatory effects on oxidative stress [79].

On the contrary, acute or chronic treadmill running at higher intensity (12 m/min speed) or downhill/uphill running generally worsened the inflammatory status of *mdx* mice [98], leading to increased infiltration of inflammatory cells [90,118] and expression for pro-inflammatory cytokines IL-1β, IL-6 [87], and TGFβ1 [81,82,83], and decreased expression of protective factors such as insulin-like growth factor 1 [89], PGC-1α and Pparγ [88], which exert anti-inflammatory actions by inhibition of NF-kB [299,312]. Moderate exercise, even in the swimming modality, is also associated to the worsening of inflammatory status, leading to the increase of inflammatory cells in cardiac muscle of 11 months old *mdx* male mice [133]. Increased inflammation and macrophages number were also observed after 3 weeks of voluntary running initiated when mice were 9 weeks old [149].

In summary, although the few available data prevent definitive conclusions, low intensity exercise and voluntary wheel running, with only few exceptions, besides improving structural and metabolic adaptions to exercise, seem to reverse oxidative stress and inflammation associated to DMD pathology, while higher intensity exercise worsens these processes. This conclusion mostly applies to limb muscles, whereas few data are available for cardiac and diaphragm muscles. The lack of an adequate amount of experimental data affects also the evaluation of *mdx* redox and inflammatory muscle state after acute training, which is usually associated to the worsening of *mdx* phenotype.

In addition to training modality, duration, intensity, and frequency as major determinants of muscle adaptation to exercise, other parameters, including age, sex, training status of the animal and time point of sacrifice following exercise, may strongly affect *mdx* muscle response to exercise. Age of exercise initiation represents a key factor in influencing muscle plastic remodeling, in both healthy and diseased subjects. DMD is a progressive chronic muscle degenerative disorder, whose symptomatology worsens with age [5]. Consequently, timing of therapy initiation, including exercise, can strongly affect the severity of clinical outcome. Accordingly, voluntary exercise is mostly associated to beneficial effects if initiated in *mdx* mice not older than 6 weeks.

DMD mainly affects boys, although female symptomatic carriers, with a different severity of DMD phenotypes, have been frequently described [313,314]. Gender, and the related hormonal variations, may greatly influence muscle physiology [315,316], disease progression and therapeutic outcome, even in response to exercise. Accordingly, female gender is associated to muscle beneficial effects in response to exercise [146] and seems to protect diaphragm from exercise-induced damage [119].

Training status of the subject, and the presence of a pre-training period and a warm-up phase associated to the main protocol of exercise, by limiting exercise-induced detrimental effects associated to sudden muscle overload, represent another key factor influencing muscle plastic remodeling in response to exercise. Indeed, pre-training and warm up, by inducing gradual metabolic adaptation to exercise and increased blood flow and temperature into the involved muscles, can decrease the risk of muscle injuries, as well as reduce heavy loads on the heart, which can occur when high intensity exercises are suddenly applied [317]. Accordingly, the presence of a pre-training phase seems to contribute to the enhancement of *mdx* muscle performance [87] and to preserve *mdx* muscles from high intensity exercise-induced damage [126,130,131].

Finally, the time of muscle sampling following exercise seems to influence the results of post-mortem analysis. Indeed, a later sampling following the last session of exercise is associated to decreased muscle damage and oxidative stress as compared to samples collected immediately after the last training session [87,113], likely due to the activation of muscle compensatory and adaptive response to exercise-dependent stress, including the activation of muscle regeneration and antioxidant defense.

The investigation of the burden and the combination of all these parameters, not enough explored in comparative studies, may
strongly contribute to explain the differences in *mdx* response to similar protocols of training and to a better understanding of pathways involved in DMD pathophysiology, as well as testing of therapeutics hypotheses, including exercise.

## 7. Conclusions and Future Perspectives

Our comparative analysis of the existing literature clearly suggests that physical exercise, by inducing beneficial muscle plastic adaptations, represents a potential not invasive therapeutic approach for improving DMD patient outcomes and quality of life.

Specifically, chronic low intensity treadmill running, at a speed less than or equal to 9 m/min, might represent the most suitable exercise modality associated to beneficial effects on *mdx* muscle, especially hind limb muscles. This typology of exercise is mostly associated to the improvement of *mdx* phenotype, including the reduction of muscle oxidative stress, inflammation and fibrosis process, and the increase in muscle functionality (force, fatigue resistance), muscle regeneration and hypertrophy (Appendix A and Figure 2). In addition, this modality of training generates reproducible results that seem to be little or not influenced by other variables, such as duration of training and age of subjects, which, on the contrary, appear to affect the outcomes of voluntary exercise, per se associated to larger individual variability.

However, when considering exercise as a therapy for DMD patients, it is essential to take into account the overall effect on patients and particularly to heart and respiratory muscles, whose failure is typically responsible of DMD patient death. Accordingly, an exercise protocol that protects or improves limb muscle function while impairing respiratory or cardiac function provides no or little therapeutic benefit to DMD patients. With respect to this key point, our analysis highlights the huge necessity for conducting new studies to deeply assess the benefits of exercise on all *mdx* muscles, including heart and diaphragm, whose functionality is poorly investigated in preclinical research. Moreover, we have shown that differences in the age, sex and training status of animals, time point of sacrifice following exercise, number of training sessions, exercise modality, intensity and frequency, might affect *mdx* muscle response to exercise, thereby pointing out the importance of new comparative studies finalized to investigate the burden of all these variables on muscle functional response to training. Given the ethical and practical issues of determining exercise prescription for DMD patients, these studies should be conducted first in dystrophic animals in order to determine functional thresholds to maximize the benefits of training and minimize the potential for exacerbating muscle injury.

These preclinical data can guide the design of appropriate studies on DMD patients. The main limitations in translational therapeutic application of *mdx* results to humans are the differences in phenotypic expression and biomechanics between humans and genetically homologous animal models. However, several studies performed in DMD patients seems to corroborate our conclusions, suggesting that submaximal low intensity exercise may be beneficial, especially if performed early in the course of the disease, while eccentric high intensity exercise is mostly associated to muscle injury.

## Figures and Tables

**Figure 1 antioxidants-10-00558-f001:**
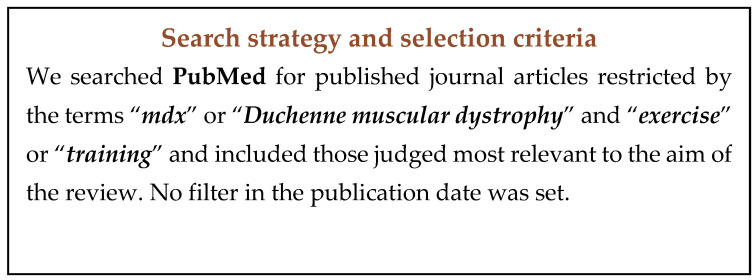
Search strategy and selection criteria of papers used in the review.

**Figure 2 antioxidants-10-00558-f002:**
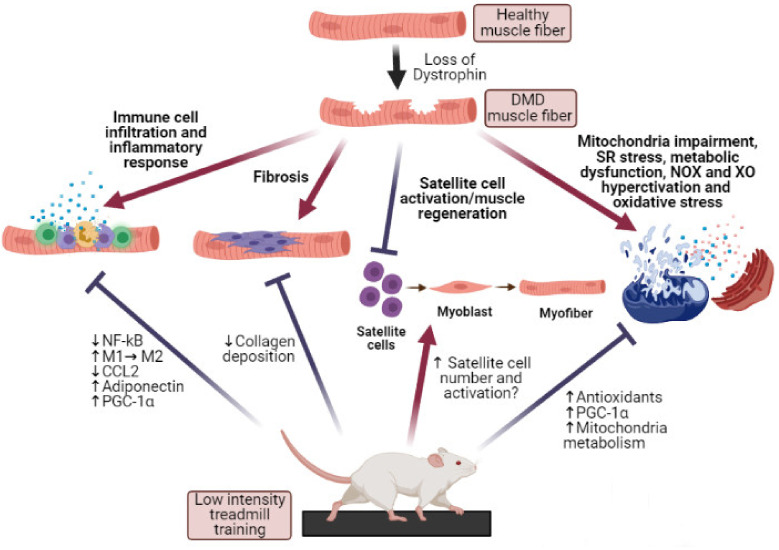
Effects of low intensity treadmill running on *mdx* muscle pathological changes. Exercise can reverse: (a) Inflammatory response, by inducing a shift of macrophage population from to M1 pro-inflammatory type to the regenerative M2 type, and by modulating inflammatory pathways and molecules, including NF-kB (Nuclear factor-kB), CCL2 (Monocyte chemoattractant protein-1), Adiponectin and PGC-1α (Peroxisome proliferator-activated receptor gamma coactivator 1-alpha); (b) Fibrosis, by reducing collagen deposition; (c) Muscle degeneration, likely by activating satellite cells; (d) Oxidative stress and metabolic impairment, and the related mitochondria and SR (sarcoplasmic reticulum) stress, NOX (NADPH oxidase) and XO (xanthine oxidase) hyperactivation, by inducing activation of antioxidants, mitochondria metabolism and PGC-1α pathway. Created with BioRender.com.

## Data Availability

Not applicable.

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
