# Peer review of "Beneficial Role of Exercise in the Modulation of mdx Muscle Plastic Remodeling and Oxidative Stress"

_antioxidants, 2021, doi:10.3390/antiox10040558_

Round 1

Reviewer 1 Report

The authors are making a genuine effort trying to summarize the effects different types of physical exercise have on mdx mice. The tables listing finding from different studies are a highlight of this manuscript.

The authors state several times in the manuscript that there are inconsistencies between different studies, meaning that very similar protocol sometime show beneficial effects, but others might have observed negative effects on muscle. The authors state that exercise type, duration, intensity, and frequency are major determinants of exercise adaptation, which is in general is correct. However, here we have very similar exercise protocols with different outcomes. The manuscript would benefit with a discussion what other parameters that could contribute to that similar protocols could lead to both beneficial/detrimental effects.

The authors include studies which has used female mdx mice which appeared to tolerate the exercise better. However, the authors need to clarify the reason for studying females in a x-linked recessive disorder.

The manuscript would appear more attractive with a schematic illustration of the pathological changes that occur in mdx muscle and what exercise could possibly counteract.

Figure 1. “Any filter in the publication date was set”. Unclear what it means, do you refer to that it could be published any year?

At some sections in the manuscript there is some misunderstandings when it comes to exercise physiology, please correct these, e.g.

  • Line 50: Hyperplasia does not happen in skeletal muscle, only hypertrophy. Also, none of the listed references 5-7 mention hyperplasia.
  • Line 80, 81: Muscle instability is not a commonly used term, please define what you mean with it.

Line 109: Which “these” do the authors refer to?

Line 134: Appears better suited with “experience” than “evidences”

Line 156-157: define high-intensity exercise as it could include so many protocols

Line 155-157: The authors first mention slow-twitch fibers but never mention fast-twitch muscle but instead introduce hypertrophy, please be consistent. Moreover, one could interpret the text as HIT exercise leads to activation of only fast-twitch fibers which is not correct as it depend on the HIT protocol used (pure sprint exercise or heavy weight lifting would be more likely to specifically activate fast-twitch fibers).

Line 161-162: Clarify that you refer to mdx muscle (exercise does generally don’t have detrimental effects on healthy muscle)

Line 179: “inclined platform” but authors talk about downhill running, thus “declined” appears more suitable.

Line 181: Please clarify why downhill running is associated with increased workload

Table 1: please define LIT

Line 217: typology appears to be the wrong type of wording in this context. “Type of exercise” appears a better alternative.

At more then one place the authors use capital letter for proteins which should not be spelled with a capital letter, please correct. (E.g Connexin, Phospholipase, Xanthine)

Mdx is sometimes written in italics and at other times not, please be consistent.

Line 252: which structural changes does the authors refer to, need to be clarified.

Line 306: please clarify what the Hermes protocol entails.

Line 329-333: Very lone sentence which makes it difficult to follow the explanation. Please simplify.

Line 395: Should be “increased muscle mass” and please specify which muscle the authors refer.

Line 396: “Shift towards a less fatigable phenotype without changes in mitochondria”. This sounds inconsistent and needs to be clarified.

Line 419: Please reword the sentence as it is either possible or not possible to draw a conclusion, not “might be possible”

Line 531: Determine or leads to?

Line 578: Although the mitochondria are the main site for ATP production which is key for contraction, the authors state that NOX accounts for most of the ROS production in exercising muscle. The authors need to strengthen argument why NOX would be activated by muscle contractions.

Line 592: Please update the information about SOD1 as the most important dismutase in skeletal muscle. The refence is from 1994 and the field has updated its views on SOD1 vs SO2 in skeletal muscle.

Line 602: Reformat reference.

Line 605: Endoplasmatic reticulum in skeletal muscle is commonly referred to as sarcoplasmatic reticulum, please clarify to what the authors refers to.

Line 638: The link between mdx and aging needs to be clarified.

Line 637-640: Seems more appropriate to focus on the beneficial effects of PGC-1a.

Author Response

We thank the reviewers for their constructive comments and suggestions, which have largely improved our manuscript. Our answers are in blu ink

Reviewer #1

The authors are making a genuine effort trying to summarize the effects different types of physical exercise have on mdx mice. The tables listing finding from different studies are a highlight of this manuscript.

We truly appreciate these positive comments.

The authors state several times in the manuscript that there are inconsistencies between different studies, meaning that very similar protocol sometime show beneficial effects, but others might have observed negative effects on muscle. The authors state that exercise type, duration, intensity, and frequency are major determinants of exercise adaptation, which is in general is correct. However, here we have very similar exercise protocols with different outcomes. The manuscript would benefit with a discussion what other parameters that could contribute to that similar protocols could lead to both beneficial/detrimental effects.

We agree with this suggestion. We have now commented more in detail the burden of other parameters in different section of the manuscript (lines 276-281, 323-332, 358-362, 427-428) and added a general discussion in lines 814-849.

The authors include studies which has used female mdx mice which appeared to tolerate the exercise better. However, the authors need to clarify the reason for studying females in a x-linked recessive disorder.

We thank the reviewer for this comment. We are aware that X-linked DMD mainly affects boys, although symptomatic carriers, with a different severity of DMD phenotypes, have been frequently described (Three cases of manifesting female carriers in patients with Duchenne muscular dystrophy. Song TJ, Lee KA, Kang SW, Cho H, Choi YC. Yonsei Med J. 2011 Jan; 52(1):192-5); Muscular and cardiac manifestations in a Duchenne-carrier harboring a dystrophin deletion of exons 12-29. Finsterer J, Stöllberger C, Freudenthaler B, Simoni D, Höftberger R, Wagner K. Intractable Rare Dis Res. 2018;7(2):120-125). Unfortunately, both male and female mdx mice have been included in several preclinical studies and grouped into one group, and very few studies have explored the impact of gender on mdx mice pathophysiology (Towards developing standard operating procedures for pre-clinical testing in the mdx mouse model of Duchenne muscular dystrophy. Grounds MD, Radley HG, Lynch GS, Nagaraju K, De Luca A. Neurobiol Dis. 2008;31:1–19). However, since gender may greatly influence muscle physiology (Gender differences in contractile and passive properties of mdx extensor digitorum longus muscle. Hakim CH, Duan D. Muscle Nerve. 2012;45(2):250-256. Gender influences cardiac function in the mdx model of Duchenne cardiomyopathy. Bostick B, Yue Y, Duan D. Muscle Nerve. 2010;42(4):600-603), disease progression and therapeutic outcome, even in response to exercise (Sex influences diaphragm muscle response in exercised mdx mice. Hermes, T.A.; Kido, L.A.; Macedo, A.B.; Mizobuti, D.S.; Moraes, L.H.R.; Somazz, M.C.; Cagnon, V.H.A.; Minatel, E. Cell Biol. Int. 2018, 42, 1611-1621), studying gender effect on mdx mice can lead to a better understanding of pathways involved in DMD pathophysiology, as well as testing of therapeutics hypothesis, including exercise. We have added a specific comment in lines 823-828.

 - The manuscript would appear more attractive with a schematic illustration of the pathological changes that occur in mdx muscle and what exercise could possibly counteract.

We agree with the reviewer. In line with this suggestion, we have now inserted a figure including the pathological changes in DMD muscles and their modulation by treadmill exercise (Fig. 1).

- Figure 1. “Any filter in the publication date was set”. Unclear what it means, do you refer to that it could be published any year?

Yes, we did not limit the range of publication dates. We have now modified the text as “No filter in the publication date was set” (Box1).

- At some sections in the manuscript there is some misunderstandings when it comes to exercise physiology, please correct these, e.g.

Line 50: Hyperplasia does not happen in skeletal muscle, only hypertrophy. Also, none of the listed references 5-7 mention hyperplasia.

We agree with the referee and we have now removed the term Hyperplasia (line 51).

- Line 80, 81: Muscle instability is not a commonly used term, please define what you mean with it.

In line with this suggestion we have specified “mechanical” (line 85).

- Line 109: Which “these” do the authors refer to?

In line with this suggestion, we have now specified the therapeutic approaches (lines 106-107).

- Line 134: Appears better suited with “experience” than “evidences”

In line with this suggestion, we have changed the text accordingly (line 132)

- Line 156-157: define high-intensity exercise as it could include so many protocols

We agree with the referee and we have now defined high-intensity exercise and provided the relative bibliography (lines 152-156).

- Line 155-157: The authors first mention slow-twitch fibers but never mention fast-twitch muscle but instead introduce hypertrophy, please be consistent. Moreover, one could interpret the text as HIT exercise leads to activation of only fast-twitch fibers which is not correct as it depend on the HIT protocol used (pure sprint exercise or heavy weight lifting would be more likely to specifically activate fast-twitch fibers).

We agree with the referee. We have now better specified the type of training (endurance and resistance), provided examples of both exercise and introduced fast-twitch fibers (lines 157-163).

- Line 161-162: Clarify that you refer to mdx muscle (exercise does generally don’t have detrimental effects on healthy muscle)

In line with this suggestion, we have now specified “in vulnerable DMD muscles” (line 167).

- Line 179: “inclined platform” but authors talk about downhill running, thus “declined” appears more suitable.

We thank the reviewer for this comment, which has allowed us to correct a few inaccuracies. We realized that we did not distinguish between downhill and uphill running in the previous version of the manuscript. We have now specified in Table 2 the slope (negative or positive) and changed the title of the table to “Effects of downhill and uphill running on mdx mouse muscles”. In the manuscript, we have now described both protocol and the main related muscle remodelling (lines 187-193) and specified “uphill” when required.

- Line 181: Please clarify why downhill running is associated with increased workload

In line with this suggestion, we have now described both downhill and uphill running muscle modifications (lines 188-193).

- Table 1: please define LIT

We think that tables are already enough long and crowded given the large amount of information included. LIT, as abbreviation, is already listed at the end of the table. Defining the protocol of LIT, MIT and HIT in all the four tables would probably complicate table reading. On the other hand, such information is well explained in the main text and all the details of training protocol are described in the tables. For a greater clarity of LIT and MIT definition in Table 1, we have now, in the abbreviation list, specified the running speed: LIT (Low Intensity Training, running speed minor than 12 m/min), MIT (Moderate Intensity Training, running speed ranging between 12 m/min and 15 m/min). The same modifications were made in Table 3.

- Line 217: typology appears to be the wrong type of wording in this context. “Type of exercise” appears a better alternative.

In line with this suggestion, we have replaced “typology” with “type” (line 224).

- At more then one place the authors use capital letter for proteins which should not be spelled with a capital letter, please correct. (E.g Connexin, Phospholipase, Xanthine)

We agree with the referee and we have corrected accordingly.

- mdx is sometimes written in italics and at other times not, please be consistent.

We agree with the referee, and we have uniformed “mdx” in italics.

- Line 252: which structural changes does the authors refer to, need to be clarified.

In line with this suggestion, we have now specified the main structural changes associated with exercise (lines 260-261).

- Line 306: please clarify what the Hermes protocol entails.

In line with this suggestion, we have now specified the details of the protocol described in Morici et al. (Ref. 76) (lines 319-320), to which the sentence refers, which in turn is similar to that used by Hermes et al. (Ref. 119), already described in the previous paragraph (lines 308-309).

- Line 329-333: Very lone sentence which makes it difficult to follow the explanation. Please simplify.

In line with this suggestion, we have simplified the sentence. We moved the second part of the sentence describing the intensity of swimming exercise to lines 344-346.

- Line 395: Should be “increased muscle mass” and please specify which muscle the authors refer.

In line with this suggestion, we have now corrected the sentence and specified the muscle type (line 391).

- Line 396: “Shift towards a less fatigable phenotype without changes in mitochondria”. This sounds inconsistent and needs to be clarified.

This is an important point and we totally agree with the referee. However, even the authors suggested the need for further studies aimed to explore this issue. In line with reviewer’ comment, we have now underlined the atypical nature of describe results (lines 393-396).

- Line 419: Please reword the sentence as it is either possible or not possible to draw a conclusion, not “might be possible”

In line with this suggestion, we have now reworded the sentence as “it is possible to conclude…” (line 421).

- Line 531: Determine or leads to?

In line to Reviewer#2’ comment, suggesting to enlarge the description of metabolism and mitochondria defects in DMD, we have rearranged many parts of the “Redox equilibrium in DMD muscles” section, with is now titled “Mitochondria impairment and redox equilibrium in DMD muscles” (line 469), including the sentence commented by the referee. However, we agree with this comment, and the correct form of the sentence would have been “leads to”.

- Line 578: Although the mitochondria are the main site for ATP production which is key for contraction, the authors state that NOX accounts for most of the ROS production in exercising muscle. The authors need to strengthen argument why NOX would be activated by muscle contractions.

In line with this comment, we have now specified why NOX is activated by muscle contractions (lines 544-546). We have also added further experimental evidence supporting the main role of NOX in ROS production during muscle contraction (lines 570-571).

- Line 592: Please update the information about SOD1 as the most important dismutase in skeletal muscle. The refence is from 1994 and the field has updated its views on SOD1 vs SO2 in skeletal muscle.

We thank the reviewer for this suggestion. Plethora of data describing the role of different isoforms of SOD in response to exercise have been reported. Now we have briefly described SOD activity in muscle, according to fiber type (lines 583-586) and modality of exercise (lines 658-662).

- Line 602: Reformat reference.

We thank the reviewer for the comment. We have now reformatted the indicated reference.

- Line 605: Endoplasmatic reticulum in skeletal muscle is commonly referred to as sarcoplasmatic reticulum, please clarify to what the authors refers to.

We apologize for the mistake. We have now replaced Endoplasmatic reticulum (ER) with Sarcoplasmatic reticulum (SR) (lines 598-615).

- Line 638: The link between mdx and aging needs to be clarified.

We thank the reviewer for this suggestion. We have now specified the link between DMD muscles, aging, and potential beneficial effects of endurance exercise (lines 641-648).

- Line 637-640: Seems more appropriate to focus on the beneficial effects of PGC-1a.

In line with this suggestion, we have briefly summarized the beneficial effects of PGC-1a activation in muscle cells (lines 631-637).

Reviewer 2 Report

The present manuscript reviews the effect of various modalities of exercise on the skeletal muscle of the mdx mouse and suggests the most effective exercise modality that could be beneficial if pursued in human patients. The review is very interesting and comprehensive but due to the extensive literature reviewed, it is quite intense to read. This could be easily improved through the addition of diagrams- I suggest at least 2-3 diagrams/figures that summarise the key points the reader should take away. For example, a summary figure of how exercise could modulate the pathological features of Duchenne Muscular Dystrophy would be useful. Moreover, the review would benefit from the addition of a section on mitochondrial/metabolism defects in Duchenne Muscular Dystrophy to balance the discussion of all topics introduced in the review. It is the one pathological feature mentioned but not discussed to the level of the other facets.

Overall, the manuscript is well written however a thorough revision of the manuscript is required to improve grammatical errors and inconsistencies throughout. I have indicated certain examples below, but general grammatical issues include that mdx is not italicised throughout, words missing here and there, extra spaces between words and abbreviations not always introduced in first instance.

Please see below for specific comments/queries regarding the manuscript.

Abstract

  • Ln 26: need to add an “s” to the end of humans

Introduction

  • The first paragraph is essentially what was in the abstract and therefore, reads repetitively. Please revise to ensure this paragraph is more distinct from the abstract

Duchenne Muscular Dystrophy

  • Small grammatical changes throughout the manuscript. For example, Ln 82, please change “to” with “with”
  • Lns 92-97: this paragraph, which is actually just a very long sentence, seems out of place here. I would suggest including it in the opening paragraph of this section to improve flow and readability
  • Lns 115-117: this sentence either needs to be included in the above paragraph or more needs to be added to it in regards to the potential benefits of exercise for DMD
  •  

The mdx model

  • Ln 120: please change the opening word “mdx” to “the mdx mouse”
  • Ln 123: please change “in details” to “particularly” or “specifically”

Effects of exercise on plastic remodelling of mdx muscle

  • Ln 179-181: please rephrase for clarity and readability
  • Table 1-4: it would be good to add a column that indicates if the study demonstrated overall positive or negative effects on the muscle
  • Table 1-4: please present the studies either in chronological or alphabetical order
  • Lns 256-259: please include this sentence with the above paragraph
  • Lns 278-282: please rephrase for readability and clarity
  • Ln 377: please replace “differently” with “different”
  • Ln 451: please change “8 or 3” to “3 or 8”

Redox equilibrium in DMD muscles

  • Ln 575: the sentence begins with “in conclusions” but the section does not conclude here. Please revise
  • Ln 602: reference not inserted correctly
  • Lns 600-604: please revise to include this sentence elsewhere

Exercise modulation of mdx muscle oxidative stress and inflammation

  • Ln 683: please change “at” to “et”
  • Ln 764: please add this paragraph to the paragraph above or revise the beginning of the sentence to be introductory

Conclusions and future perspectives

  • Lns 804-806: please add this sentence to the proceeding paragraph

Author Response

We thank the reviewers for their constructive comments and suggestions, which have largely improved our manuscript. Our answers are in blu ink

Reviewer #2

- The present manuscript reviews the effect of various modalities of exercise on the skeletal muscle of the mdx mouse and suggests the most effective exercise modality that could be beneficial if pursued in human patients. The review is very interesting and comprehensive but due to the extensive literature reviewed, it is quite intense to read.

We thank the reviewer for the appreciation.

- This could be easily improved through the addition of diagrams- I suggest at least 2-3 diagrams/figures that summarise the key points the reader should take away. For example, a summary figure of how exercise could modulate the pathological features of Duchenne Muscular Dystrophy would be useful.

In line with this suggestion, we have now inserted, at the end of the manuscript, a figure including the pathological changes in DMD muscles and their modulation by exercise (Fig. 1).

- Moreover, the review would benefit from the addition of a section on mitochondrial/metabolism defects in Duchenne Muscular Dystrophy to balance the discussion of all topics introduced in the review. It is the one pathological feature mentioned but not discussed to the level of the other facets.

In line with this comment we have now introduced, at the beginning of “Redox equilibrium in DMD muscles” section, a new paragraph briefly describing the mitochondrial/metabolism defects in Duchenne Muscular Dystrophy (lines 470-476, 505-519). The title of the section has been changed to “Mitochondria impairment and redox equilibrium in DMD muscles” (line 469).

- Overall, the manuscript is well written however a thorough revision of the manuscript is required to improve grammatical errors and inconsistencies throughout. I have indicated certain examples below, but general grammatical issues include that mdx is not italicised throughout, words missing here and there, extra spaces between words and abbreviations not always introduced in first instance.

We thank the reviewer for the appreciation. We apologize for grammatical errors and text formatting issues. We have now carefully checked the text and made the relative corrections.

Please see below for specific comments/queries regarding the manuscript.

Abstract

- Ln 26: need to add an “s” to the end of humans

In line with this suggestion, we have changed the text accordingly (line 21).

Introduction

The first paragraph is essentially what was in the abstract and therefore, reads repetitively. Please revise to ensure this paragraph is more distinct from the abstract

We agree with the reviewer and we have revised the text according to his suggestion (lines 34- 44).

Duchenne Muscular Dystrophy

- Small grammatical changes throughout the manuscript. For example, Ln 82, please change “to” with “with”

We apologize for the grammatical error. We have now corrected the text accordingly (line 85).

- Lns 92-97: this paragraph, which is actually just a very long sentence, seems out of place here. I would suggest including it in the opening paragraph of this section to improve flow and readability

In line with this comment, we have changed the text accordingly. All the information has been moved and included in lines 74-84.

- Lns 115-117: this sentence either needs to be included in the above paragraph or more needs to be added to it in regards to the potential benefits of exercise for DMD

In line with this suggestion, we have added new information to the sentence and moved it to the above paragraph (lines 112-114).

The mdx model

- Ln 120: please change the opening word “ mdx” to “the mdx mouse”

In line with this suggestion, we have modified the text accordingly (line 117).

- Ln 123: please change “in details” to “particularly” or “specifically”

In line with this suggestion, we have now replaced “in details” with “specifically” (line 120).

Effects of exercise on plastic remodelling of mdx muscle

- Ln 179-181: please rephrase for clarity and readability

In line with this suggestion and reviewer#1’ comment, we have updated and rephrased the sentence (lines 187-193).

- Table 1-4: it would be good to add a column that indicates if the study demonstrated overall positive or negative effects on the muscle

In line with this suggestion, we have added a column indicating the effects of the protocol of exercise on mdx muscles: (+) Beneficial effects, (±) No effects or discordant effects; (-) Detrimental effects (see Table 1-4).

- Table 1-4: please present the studies either in chronological or alphabetical order

We agree with the reviewer and we have now listed the studies in chronological order, but maintaining the internal division in LIT, MIT and HIT. We think that this subdivision in the presentation of the studies may help in table reading and data interpretation.

- Lns 256-259: please include this sentence with the above paragraph

In line with this comment, we have moved the sentence to the above paragraph (line 264).

- Lns 278-282: please rephrase for readability and clarity

In line with this suggestion, we have rephrased accordingly (lines 291-294).

- Ln 377: please replace “differently” with “different”

We have changed the text according to this suggestion (line 373).

Ln 451: please change “8 or 3” to “3 or 8”

We agree with the reviewer, and we have changed the text accordingly (line 453).

Redox equilibrium in DMD muscles

- Ln 575: the sentence begins with “in conclusions” but the section does not conclude here. Please revise

We agree with the reviewer, and we have now replaced “conclusion” with “summary” (line 565).

- Ln 602: reference not inserted correctly

We thank the reviewer for the comment. We have now reformatted the indicated reference (Ref. 181) (line 597).

- Lns 600-604: please revise to include this sentence elsewhere

In agreement with this comment, we have rephrased the sentence, which has been moved to the previous paragraph (lines 594-597).

Exercise modulation of mdx muscle oxidative stress and inflammation

- Ln 683: please change “at” to “et”

We apologize for the mistake, which has been now corrected (line 694).

- Ln 764: please add this paragraph to the paragraph above or revise the beginning of the sentence to be introductory

In line with this suggestion, we have moved the paragraph to the previous one (line 774).

Conclusions and future perspectives

- Lns 804-806: please add this sentence to the proceeding paragraph

In line with this suggestion, we have changed the text accordingly (lines 852-854).

Round 2

Reviewer 1 Report

In the revised version of the manuscipt, the authors have carefully responded and addressed the concerns that were raised. No additional comments, but please change "Peroxisome" to peroxisome on line 530.  

Reviewer 2 Report

Dear Authors,

Thank you for addressing my comments and implementing several of my suggestions.